# A Post-Evaluation System for Smart Grids Based on Microservice Framework and Big Data Analysis

Jie Wang [1], Ruiqi Ouyang [1], Wu Wen [1,*], Xin Wan [1], Wei Wang [2], Amr Tolba [3,*] and Xingguo Zhang [4]

1   School of Communication and Information Engineering, Chongqing University of Posts and Telecommunications, Chongqing 400065, China
2   School of Software, Dalian University of Technology, Dalian 116024, China
3   Department of Computer Science, Community College, King Saud University, Riyadh 11437, Saudi Arabia
4   Department of Mechanical Systems Engineering, Tokyo University of Agriculture and Technology, Nakacho Koganei, Tokyo 184-8588, Japan
*   Correspondence: wenwu@cqupt.edu.cn (W.W.); atolba@ksu.edu.sa (A.T.)

**Abstract:** Wind energy, as a clean energy source, has been experiencing rapid development in recent years. However, there is often a significant difference between the designed electricity generation capacity and the actual electricity generation capacity during the construction of wind farms, making it difficult to assess the economic benefits of wind farms. Therefore, the development post-evaluation technology is required to support the renovation of old wind farms. In addition, traditional data analysis techniques are unable to handle and analyze massive data in a timely manner. Therefore, big data technology must be developed to improve processing efficiency. To address these issues and meet actual business needs, this paper designs an intelligent grid electricity generation post-evaluation platform for wind farms based on a microservice framework and big data analysis technology. The platform evaluates the operating status of wind farms by analyzing their operational and design data and visualizes relevant big data information. It provides technical support and improvement solutions for wind farm renovation and new wind farm construction. The platform has been tested and proven to meet the requirements for processing and analyzing massive data, post-evaluating electricity generation, and visualization.

**Keywords:** micro-services framework; grid big data assessment; supervisory control and data acquisition; nacelle transfer function; smart grid

## 1. Introduction

In recent years, due to the growing global demand for energy and increasing awareness of the need for environmental protection, wind energy has become a popular clean energy source. More and more countries and regions are investing in wind farms to meet their energy needs and promote sustainable development. However, as the number of wind farms continues to increase, the efficient management and monitoring of these distributed wind power facilities have become major challenges. These wind farms usually consist of hundreds of wind turbines, each of which needs to be monitored and maintained. Additionally, the instability of wind energy also makes wind farm management more challenging, as changes in wind speed and direction can have a significant impact on the performance and output of wind turbines.

In this context, big data technology can play an important role. By monitoring and collecting large amounts of wind farm data, such as wind speed, wind direction, turbine output, and power load, management personnel can better understand the operational status of wind farms. In addition, big data technology can also be used for making predictions and optimizing wind farms, such as optimizing turbine output by predicting changes in weather and wind speed and adjusting the load balance of the power grid.

### 1.1. Motivation

Post-evaluation of wind farm power generation is the process of evaluating the performance of wind turbines and determining their electricity generation capabilities. With the increasing number and size of wind farms, the demand for power post-evaluation has also been on the rise. In the current era of rapid technological development, it is necessary to continuously apply new technologies to practical business matters and innovate continuously based on this principle to maintain a competitive advantage in the wind power industry. However, wind farm researchers currently only rely on manual analysis software to evaluate a large amount of wind farm data. As data volumes increase and delay-sensitive applications continue to develop, the need for ubiquitous connectivity and high-accuracy analysis continues to rise, and traditional manual analysis methods have become inefficient. In this context, it is necessary to develop a digital platform for post-evaluation of wind farm power generation based on big data technology in order to promote the modernization and intelligence of wind farm platforms and complete the overall planning of enterprise wind farms.

### 1.2. Research Challenge

The continuous development of big data technology has brought new opportunities for assessing wind farm power output. The application of big data technology to evaluate wind farm power generation post-operation can enable more precise assessments of wind turbine performance and output, thereby improving the operational efficiency and power generation capacity of wind farms. However, such assessments typically involve a large amount of data processing and analysis, and thus pose certain challenges.

- Incomplete or missing data: In the post-evaluation of wind farm power, the collected data may be incomplete or missing, which can reduce the accuracy and reliability of the assessment results.
- Inconsistent data quality: The data generated during the operation of wind turbines may contain noise or errors, which can impact the accuracy of data analysis. Additionally, due to the distributed nature of wind turbines, data may originate from different sources, each of which may have different standards for data quality.
- Difficult to establish accurate benchmarks: To accurately determine the performance of wind turbines, it is necessary to compare them with data from other similar turbines. However, establishing accurate benchmarks can be challenging due to differences in turbine models, environmental conditions, and service life.
- Difficult to predict future power generation: Environmental factors such as wind speed and direction can affect the performance and output of wind turbines, making the evaluation of their future power generation capacity uncertain.
- Large-scale data analysis and processing: Wind farms generate a vast amount of data, which requires a large amount of analysis and processing to extract valuable information. This necessitates the use of advanced algorithms and analytical techniques to process and analyze a large volume of data.

### 1.3. Contributions

This article presents a platform for the post-evaluation of wind farm power generation, which utilizes big data processing and analysis techniques along with a distributed software architecture to address relevant issues. The main contributions of this paper are as follows:

- This study employs big data processing technology to enhance the efficiency of post-evaluation computation and processing while integrating specific post-evaluation business content related to wind farm power generation. To ensure the professionalism and reliability of post-evaluation assessments, the study utilizes monitoring and control mechanisms, data acquisition, and nacelle transfer functions for the quantitative evaluation of power generation.
- The system incorporates a distributed software architecture. This can significantly enhance the overall throughput of the system compared to traditional centralized

application systems, thereby reducing system coupling and latency. This results in efficient system operation, improved system reliability, and greater system stability.

- The system incorporates big data visualization technology into the smart grid to achieve the visualization of massive data processing results. The use of big data processing results in more intuitive and readable calculation outcomes. This aids wind resource engineers in making informed judgments about the operational status of wind farms, thus improving the operating efficiency and power generation efficiency of the system.

- This article presents a comprehensive solution based on big data for the construction of a post-evaluation system for power generation in wind farms. The system described in this article offers valuable insights and references for the creation of future post-evaluation platforms for power generation based on big data.

The remainder of this article is organized as follows: Section 2 provides a summary of related research on smart grids and associated fields. Section 3 presents the system design for post-evaluating wind resources in wind farms. Section 4 details the system's implementation. Section 5 discusses the system testing and presents the resulting data. Finally, Section 6 provides a discussion and summary of the article.

## 2. Related Work

In the past, software applications for intelligent grid power generation assessment received little attention, and many data analysis and processing tasks were performed by individual engineers using primitive methods such as Excel, which were not only time-consuming but also prone to errors. However, with the emergence of automation technology, intelligent grid power generation assessment platforms have adopted automation technologies, such as automatic meter-reading and smart meter technologies, to reduce manual data entry and improve data accuracy and real-time performance [1]. Currently, intelligent grid power generation assessment platforms rely heavily on intelligent solutions based on power grid big data, cloud computing, and artificial intelligence technologies. These platforms can monitor and analyze power grid data in real-time, detect abnormal situations and problems, and provide predictions and warnings [2,3]. Furthermore, these platforms have enhanced their data processing and analysis capabilities, leading to more scientific and accurate evaluation results. At this stage, assessment platforms also support visual data presentation and interactive analysis tools to facilitate user understanding and utilization of evaluation results, leading to better management and optimization of the power grid.

The McKinsey Global Institute (MGI) [4] proposed the revolutionary concept of "electric power big data" in 2011. The ultimate goal of electric power big data is to achieve innovative patterns and application improvements for typical business scenarios. By employing key technologies such as data integration management, storage, computation, analysis, and mining, electric power big data can facilitate business trend prediction and data value mining. Researchers [5] have investigated the relationship between big data, cloud computing, and smart grids and have presented a comprehensive framework for electric power big data platforms in the literature. Mayer [6] underscores the critical importance of electric power big data for smart grids. Through the transmission of vital information such as users' electricity consumption habits to the information center of electric power companies, network analysis and processing can significantly impact the planning, construction, and service aspects of the electric power grid.

In the realm of electric power, the application of big data is not solely a technological advancement, but also involves significant changes in the development concept, management system, and technical route of the entire power system in the era of big data. This transformative shift represents a leap in the value-form of the next generation of intelligent power systems in the era of big data [7]. Smart grids offer numerous advantages over traditional power grids by integrating the production, transmission, distribution, and safety protection of electricity with advanced information technology [8,9]. The large-scale smart

grids of the future will operate on the energy internet, with data sets sent along power routers to specific destinations. To address the challenge of large-scale computation in smart grids, Hou et al. [10] have designed a spatiotemporal big data computation framework for large-scale smart grids to improve computation efficiency and save path bandwidth. Moreover, through data analysis and artificial intelligence algorithms, prediction and recommendation functions can be achieved [11]. For instance, in reference [12], a neural network was utilized to construct a prediction model for generator bearing temperature, with the component state being judged based on the deviation between the actual value and the predicted value. Reference [13] completed the prediction of wind turbine output power by combining various machine learning algorithms, and proposed the use of an edge AI-based prediction framework to enhance the efficiency of intelligent micro-grids. To improve the accuracy of wind power prediction, Lv et al. [14] combined different deep learning algorithms with edge computing [15,16] to analyze and process the distributed renewable energy generation and consumer power data in intelligent micro-grids, thereby improving information transmission and processing efficiency in the power system.

The applications of big data in the electricity industry involve two central themes: reshaping the fundamental value of electricity and transforming its development. For traditional centralized power grid systems, integrating and coordinating a vast and continually expanding number of connections can present significant challenges. Consequently, smart grids are shifting from their centralized configuration to a decentralized topology. Furthermore, research related to big data mainly focuses on combining distributed technology [17] and edge computing to achieve decentralization. Distributed computing can boost computation efficiency, enhance overall system performance, provide computing resources to end-users, and ensure low latency. To achieve the intelligent allocation and scheduling of distributed computing resources and edge computing resources [18], Kong et al. [19] considered simultaneously the energy cost of computation and caching. They used deep reinforcement learning to minimize the energy cost of mobile network operators. In distributed computing tasks, the distributed alternating direction method of multipliers algorithm [20] can select task computing modes in a distributed manner in order to benefit the industry. While distributed systems bring convenience, they also come with corresponding security issues. To address security issues in distributed systems, Ning [21], Gai [22], Wang [23], and others have combined edge computing, distributed technology, and blockchain technology to optimize data security, user utility, and system latency under limited resources.

The control system is an automated system designed to remotely monitor and control wind turbines. As a vital component of the intelligent grid assessment platform, the monitoring system is computer network-based and capable of achieving functions like remote data collection and regional management. In theoretical research, Huang et al. [24] used archived customer instrument intervals and supervisory control and data acquisition (SCADA) with the same timestamp to construct quasi-dynamic models offline and predict the state and measurement values of feeder busbars, thus enhancing the system operation model and network efficiency. The power big data platform is the practical application of IoT technology, big data concepts, technologies, and methods in the power industry [25,26]. Big data covers all aspects of power generation, transmission, transformation, distribution, consumption, and scheduling [27], and involves the collection of massive business data generated from power monitoring, production operation, marketing management, and customer service. It requires cross-unit, cross-disciplinary, and cross-business data collection [28], storage, management, analysis, and visualization. The introduction of big data technology into smart grids [29] can make the grid system more intelligent and efficient [30].

## 3. System Design

### 3.1. Requirements Analysis

The main function of this platform is to analyze and calculate the operating conditions of existing wind farms, monitor the distribution of wind farms nationwide, and evaluate

their operating conditions by comparing statistical indicators of different wind farms. The operating conditions of individual wind farms are evaluated by analyzing their unit, wind measurement tower, and fault data. Corresponding wind resource assessment reports are issued based on this information. To meet these requirements, the system is divided into project module, monitoring and control module, data collection module, wind measurement tower module, nacelle transfer function module, operation analysis module, model validation module, and report module. The project module is used to manage projects created by users, such as searching, modifying, creating, and deleting by region. It also displays evaluation indicators for individual wind farms, such as the power-based approach (PBA) score and discount score.

The monitoring and data acquisition module is designed to perform statistical analysis on the raw measurement data of wind turbines, assess the quality of the data, and analyze its impact on subsequent calculation results. The wind measurement tower module is used to display the operational data of individual wind measurement towers. The nacelle transfer function module selects an appropriate nacelle transfer function by considering various wind measurement towers and units. The operation analysis module displays the wind farm's operational status after processing the raw data, including power generation, PBA, average wind speed, and other indicators. The model verification module compares the design and actual operational conditions, analyzes the differences between the two periods, and identifies the reasons for those differences.

The report module is used to publish corresponding evaluation reports based on the operating conditions of the wind farm.

- Security

Security is a necessary requirement for any system. Without security, a system cannot be put into use. This responsibility not only lies with the software itself, but also with the users who use it. To ensure high security, the entire system is deployed on Amazon servers, which ensures the high availability of its services. Additionally, since the system is currently only used within the company, access permissions are restricted to the internal network to avoid any external attacks. At the code level, measures such as gateway interception and token verification are implemented to ensure the security of interfaces and data.

- Computational accuracy

The system is designed for wind resource engineers and does not require high concurrency computing. Therefore, the main focus is on ensuring accurate calculation, differentiated service [31], and the efficient allocation of network resources [32]. Since each calculation program takes a long time and consumes many resources, the system only needs to ensure that correct data produce accurate calculation results without considering real-time performance [33] and service performance, which is not feasible. To decouple computation from the system, a message queue is used to notify the back-end to perform the calculation. The back-end completes the specific calculation, and the result of the calculation success or failure is returned after the back-end completes the calculation.

- Scalability

To ensure the sustainable development of the system, iteration and secondary development are essential processes. In these processes, ensuring quick deployment and program availability to improve network performance and provide high-quality services to users is a critical issue that must be considered [34]. To ensure the scalability of the system, a micro-service architecture has been adopted. The back-end has been encapsulated into different API, which are provided to the front-end to enable calling according to the module. At the same time, a circuit breaker mechanism and a load balancer have also been added to the system. All API have been deployed on Docker, which means that when a service has a problem, it will not affect the normal operation of other services. Additionally, using Docker is also conducive to service deployment and version management. During the iteration process, new services can be developed and deployed based on the original services.

### 3.2. System Function and Overall Design

#### 3.2.1. Function Module Design

According to the actual functional requirements, the system modules are divided as shown in Figure 1.

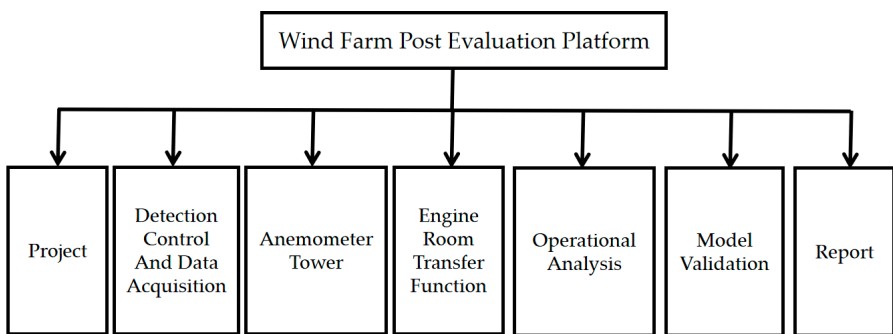

**Figure 1.** Function Module Diagram.

This project comprises four modules: post-assessment project management, list display, project creation, and project overview. The project management module shows the geographic distribution of wind farms across the country, allowing users to view wind farms by region and owner. The project list displays information such as project name, address, owner information, installed capacity, turbine model, project creator, and more kinds. The project creation module allows users to input project information and upload wind farm operation data. The project overview module displays important indicators after the wind farm assessment is completed.

The SCADA module is used for analyzing and processing the raw operating data of turbines, which include wind direction normalization, quality statistics, and interpolation. The met mast module displays the operating data of the met mast in the wind farm, which are divided into four groups: overview, time-series plot, parameter calculation, and statistics.

The Nacelle transfer function (NTF) module is the core calculation module of the entire platform, which includes NTF calculation, NTF check, and NTF application. The NTF calculation includes displaying the thermodynamic map of the turbine cabin, comparing the wind speed time-series plots between the cabin and the met mast, and the distribution of the free flow sector of the met mast and matching turbine. The NTF application includes comparing the cabin wind speed and the corrected wind speed. The operating analysis module displays the operating status of the wind farm, which includes five parts: overview, power curve, PBA, control strategy viewing, and fault analysis.

The model verification module is designed to compare pre-design and post-operation data in order to view and analyze the differences between the two periods. It is divided into two parts: pre–post difference comparison and pre–post difference analysis. The pre–post difference comparison includes a comparison of power generation hours and average wind speed, wind frequency and corresponding electricity production, as well as the reduction factor of electricity generation and the software simulation error between the two periods. The pre–post difference analysis includes representative year analysis, met mast representativeness analysis, and power generation deviation analysis.

The operation data are calculated and analyzed to determine the operating status of the wind farm and generate an evaluation report. Users can read the report to obtain information on the wind farm's operating status.

#### 3.2.2. System Architecture Design

To facilitate the rapid development of this project, we adopted a front-end and back-end separation architecture. The front-end and back-end are independently developed based on predefined interfaces, and debugging is performed after both the front-end and

back-end of a certain module are completed. This is followed by the debugging of the entire project. The architecture of the system is shown in Figure 2 [35].

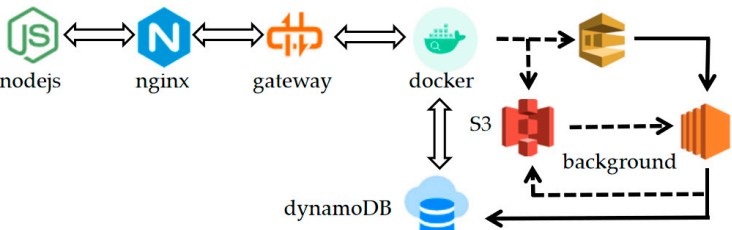

**Figure 2.** System Architecture.

To achieve the separation and decoupling of the front-end and back-end, the system adopts a front-end and back-end separation architecture pattern. The front-end uses technologies such as h5 [36], Echart [37], and Node.js [38] to access the system, transmit data, and display, beautify, and render results by accessing predefined API interfaces. To ensure the stability and consistency of the interfaces, all API developed by the back-end are forwarded by NGINX [39]. All requests are received and processed by the gateway layer, and only requests that meet the requirements are sent to the API layer to be accessed and calculated. To ensure the decoupling and independent development of modules, each module is rendered as a service, and all services are deployed on Docker [40] for ease of version iteration and redeployment.

Most API only involve data creation, retrieval, updating, and deletion, and they directly access DynamoDB [41] for the corresponding operations. Some API involve file transmission and computation, and these API need to store files in a simple storage service (S3) or send messages to a simple queue service (SQS) [42]. The back-end receives and consumes these messages, and launches different calculation scripts according to the type and content of the message, finally writing the calculation results into DynamoDB and S3 and notifying the system when the computation is completed.

3.2.3. System Storage Structure Design

The system involves a large number of computational files. To ensure the security and availability of these files, all files are stored in Amazon S3, and the storage directory structure is shown in Figure 3.

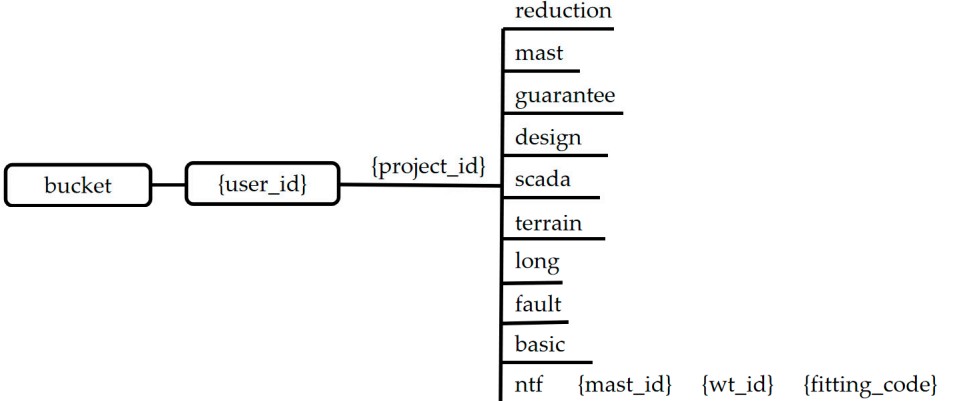

**Figure 3.** S3 Storage Structure.

The "Bucket" is the storage container in S3, "user_id" represents the user's ID, and all project data files for that user are stored in this directory. "project_id" represents the project's ID, and all calculated input and output data files for the project are stored in this directory. The "terrain" directory stores the topography files, the "long" directory

stores the long-term wind speed files, the "fault" directory stores the fault data files, the "basic" directory stores the basic information files, and the "NTF" directory stores the SCADA-corrected files generated by the NTF application.

To achieve system decoupling, a message queue is designed to send messages from the API and process them in the background. As there are seven calculations, seven message queues have been designed, with each queue responsible for completing a specific task. The seven message queues are: wind tower calculation, SCADA pre-processing, SCADA interpolation, NTF pre-processing, NTF application, operational analysis and model verification, and terrain verification.

## 4. System Implementation

Due to the adoption of a front-end and back-end separation architecture, the programming languages used for the front-end and back-end do depend on each other, but instead are determined based on the system's design and implementation. As the system involves a large amount of chart display, the front-end uses the h5 + Echart style to achieve optimal rendering effects. Additionally, the front-end and back-end interact through restful interfaces named by Node.js.

### 4.1. System Front-End

The system front-end is mainly divided into the following modules:

- Project Module

The project module, shown in Figure 4, contains all basic information related to the project. The project overview page displays the overview of all projects, where users can view, create, delete, edit, and calculate project information. This module covers the geographic distribution and four statistical indicators of a wind farm in a specific region. These include wind speed–power generation statistics, PBA–curtailment rate statistics, actual curtailment coefficient statistics, and software simulation error statistics. Apart from these four statistical indicators, the module also includes geographic distribution information of the wind farm in that region. The map provides a visual layout of the wind farm and the position of each wind turbine. Users can zoom and drag the map as per their requirements to better understand the geographic distribution of the wind farm.

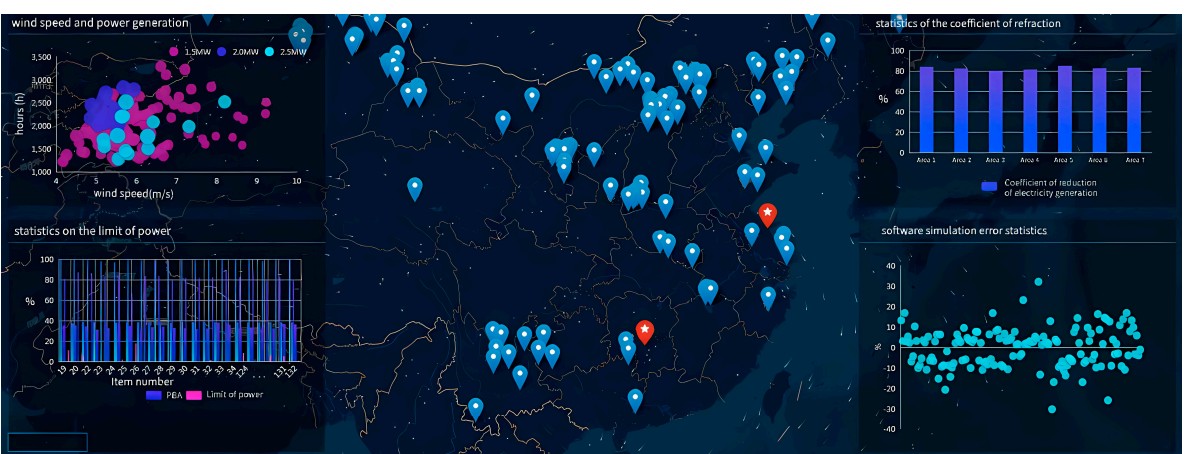

**Figure 4.** Project Module.

- Detection Control and Data Acquisition Module

The interface of the detection control and data acquisition module is shown in Figure 5 and is primarily used for the statistical analysis of raw data from wind turbines to verify their rationality and effectiveness, ensuring the accuracy of calculation files. This module includes a wind direction normalization page, which is used to display the comparison information from before and after wind direction normalization. In a wind farm, due to

the close distance between different turbine locations, wind speed and wind direction are almost the same. However, inaccurate wind direction measurement instruments in turbines can cause a large difference in the wind direction between them, making them unsuitable for statistical analysis of the wind direction time series in the wind farm. Therefore, wind direction normalization must be performed to correct the wind direction of all turbines to the same relative position. After correction, the trend of the time series tends to be consistent. If the trend is inconsistent after correction, it may be due to problems with the input data, and the input data should be checked; otherwise, it may cause calculation failure later on.

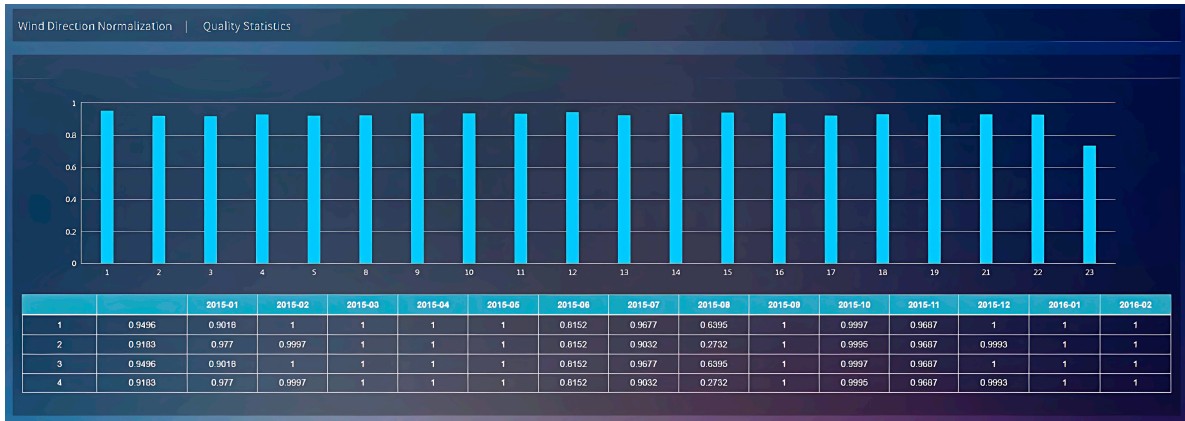

| | | 2015-01 | 2015-02 | 2015-03 | 2015-04 | 2015-05 | 2015-06 | 2015-07 | 2015-08 | 2015-09 | 2015-10 | 2015-11 | 2015-12 | 2016-01 | 2016-02 |
|---|---|---|---|---|---|---|---|---|---|---|---|---|---|---|---|
| 1 | 0.9496 | 0.9018 | 1 | 1 | 1 | 1 | 0.8152 | 0.9677 | 0.6395 | 1 | 0.9997 | 0.9687 | 1 | 1 | 1 |
| 2 | 0.9183 | 0.977 | 0.9997 | 1 | 1 | 1 | 0.8152 | 0.9032 | 0.2732 | 1 | 0.9995 | 0.9687 | 0.9993 | 1 | 1 |
| 3 | 0.9496 | 0.9018 | 1 | 1 | 1 | 1 | 0.8152 | 0.9677 | 0.6395 | 1 | 0.9997 | 0.9687 | 1 | 1 | 1 |
| 4 | 0.9183 | 0.977 | 0.9997 | 1 | 1 | 1 | 0.8152 | 0.9032 | 0.2732 | 1 | 0.9995 | 0.9687 | 0.9993 | 1 | 1 |

**Figure 5.** Detection Control and Data Acquisition Module.

This module also includes a quality statistics page, which includes the overall data completeness rate of the unit, the monthly data completeness rate of the unit, and statistics of wind speed, wind direction, and rotational speed of the unit. Due to equipment failures, extreme weather, data transmission loss, and other reasons, a certain amount of measurement time series will be lost, which is acceptable in actual calculations. When the actual amount of data reaches or exceeds 90% of the theoretical amount of data, it can be considered that the result will not have a significant impact on the calculation result.

- Wind Measurement Tower Module

The user interface of the wind measurement tower module is shown in Figure 6, which includes basic information on the operation of the tower. This module provides a wind shear diagram showing the variation in wind speed with the measurement height, a 16-wind-rose diagram for different height levels (showing the frequency of data statistics in each directional range), a graph showing the variation in wind speed with month, and a graph showing the variation in wind speed with time. In addition, statistical results for air density and temperature variation per month are also provided. By viewing these statistical results, users can understand the operation of the wind measurement tower.

- Cabin Transfer Function Module

The cabin transfer function module is shown in Figure 7. The main function of this module is to match the time series of cabin wind speed and wind tower wind speed, calculate the NTF parameters, and correct and calculate the wind measurement data. Then, suitable cabin transfer functions are used for operational analysis and model validation. This module displays the information processed by NTF, including the comparison of wind speed–time series between selected units and wind towers, as well as the free flow fan area. In addition, scatter plots and fitting curves of cabin wind speed and wind tower wind speed during operation and shutdown are also displayed.

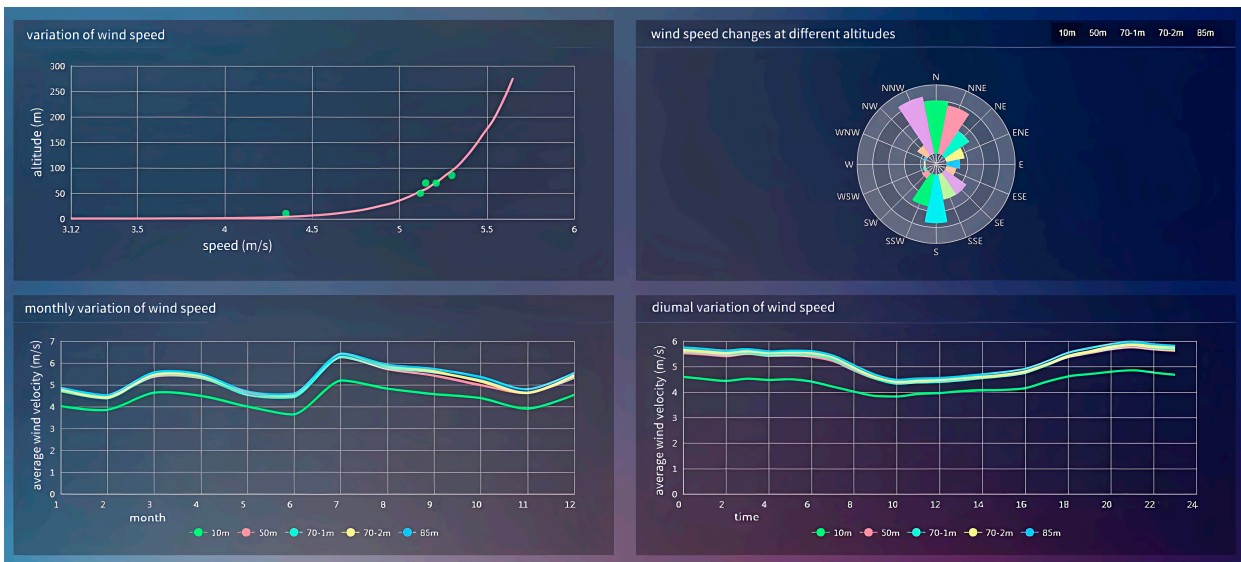

**Figure 6.** Wind Measurement Tower Module.

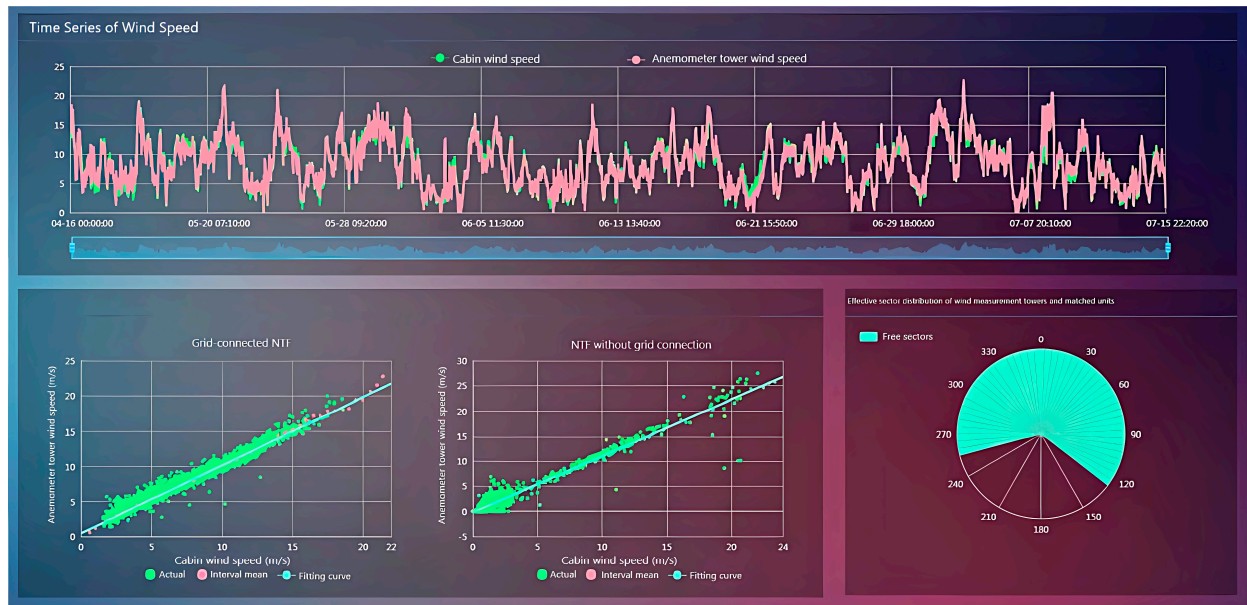

**Figure 7.** Cabin Transfer Function Module.

- Operation Analysis Module

The operational analysis module, as shown in Figure 8, covers the overall operational status of the wind farm, including wind speed statistics, power generation statistics, fault statistics, and more. Specifically, the operational analysis overview displays the power generation hours and wind speed statistics of the wind farm. This includes a comparison of the annual average wind speed and annual power generation hours of a single unit, a monthly wind speed and power generation comparison, a contribution of power generation hours in each wind speed interval, the determination of cumulative power generation hours in each wind speed interval, and the wind frequency distribution and turbulence changes of a single unit.

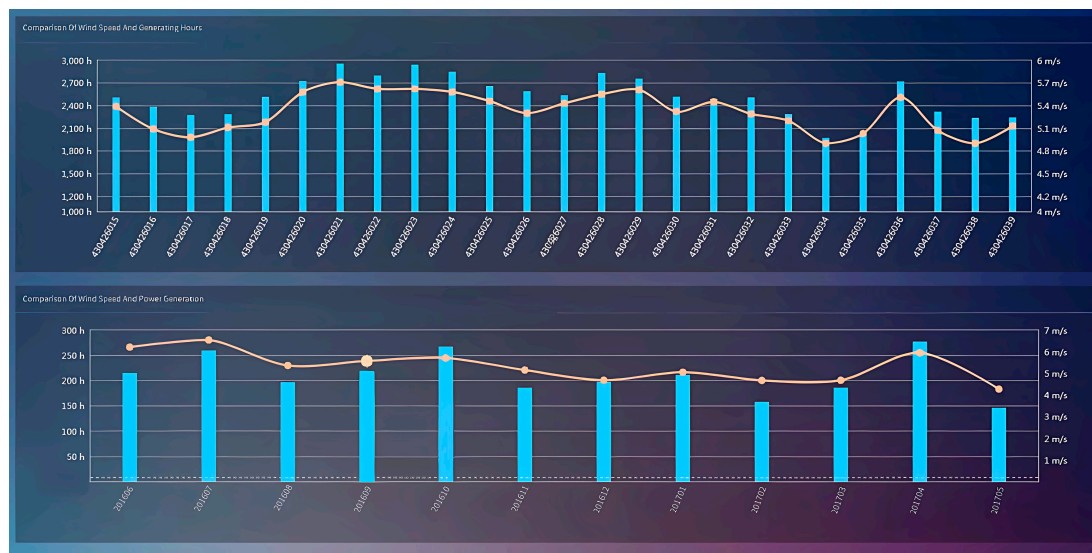

**Figure 8.** Operation Analysis Module.

- Model Validation Module

The model validation module, as shown in Figure 9, is mainly used to compare the preliminary design data with actual operational data, check the conformity between the design and operation, and analyze the reasons for the differences. This module includes a comparison of the pre- and post-construction hours and average wind speeds, a comparison of the pre- and post-construction wind frequency and corresponding power generation, a comparison of the power generation reduction coefficient, and software simulation errors. Specifically, the difference analysis between the pre- and post-construction periods shows the representative annual analysis, representative analysis of the wind measurement tower, and analysis of power generation deviation. The pre-construction reduction items include wake correction, air density correction, control and turbulence reduction, blade contamination reduction, unit availability reduction, field electricity and line loss reduction, climate-induced downtime reduction, uncertainty, and other power generation losses.

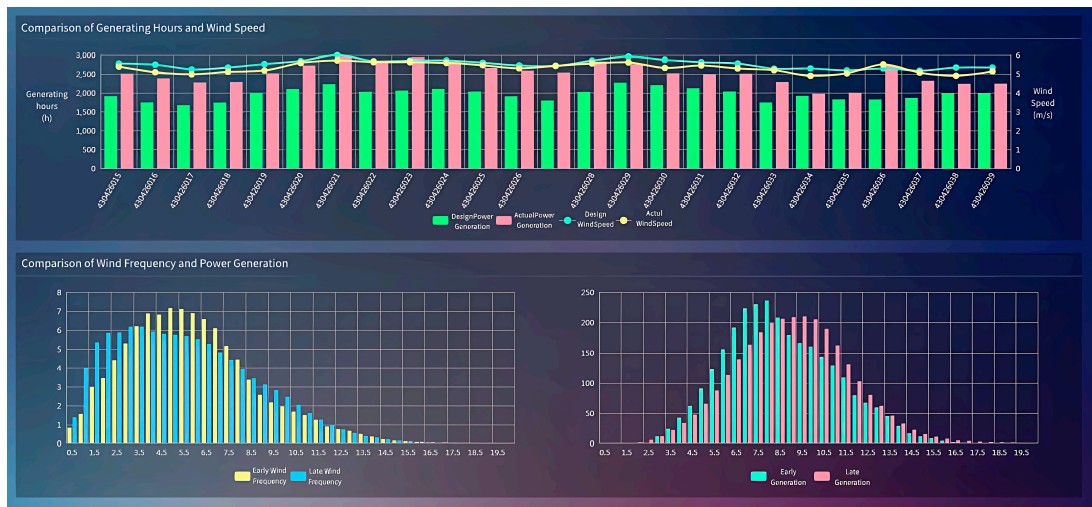

**Figure 9.** Model Validation Module.

- Report module

After completing a project, the system will publish a corresponding post-evaluation report based on the project's operation status. The report includes an introduction, basic

project information, reference standards, data collection checklist, SCADA data statistics, wind measurement tower evaluation, and operational analysis assessment. The report can be viewed online or downloaded locally. The report module will be based on the six basic modules mentioned above for analysis, and the specific process is shown in Figure 10.

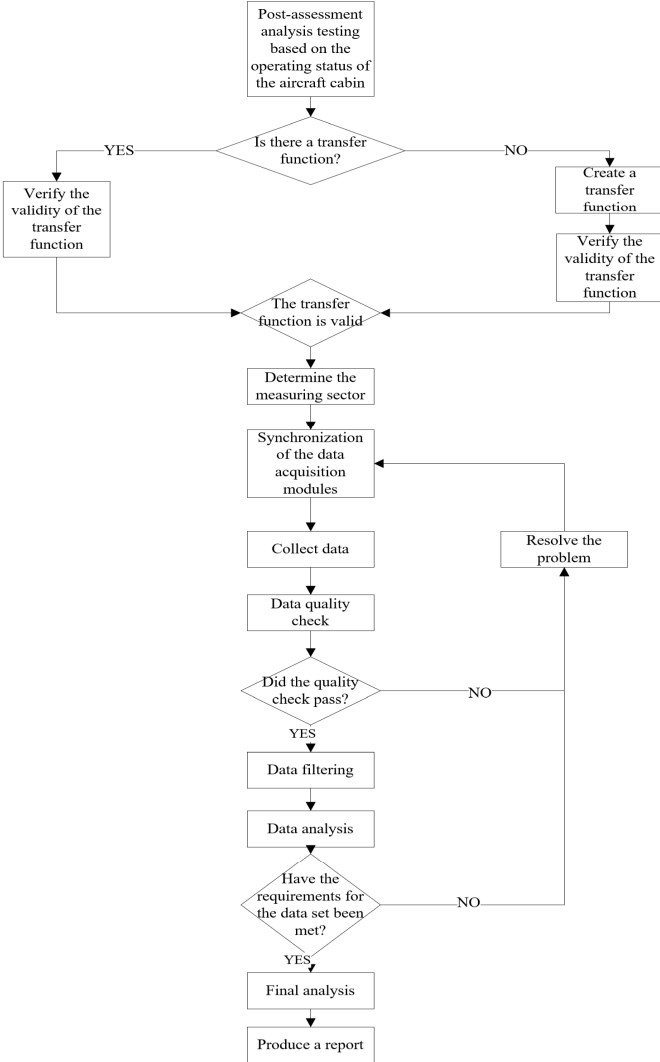

**Figure 10.** Report Module.

### 4.2. System Back-End

The main role of the back-end is to receive messages sent by the API and perform tasks such as downloading and verifying files, invoking calculation scripts, and writing the calculation results into a database, depending on the requirements. These calculations involve diverse file formats and complex logical judgments, requiring scripts written in languages such as Python and R [43]. Thus, multiple languages need to be combined and SQS messages polled. Therefore, in practical development, we used the Spring Boot framework in Java.

The back-end structure is shown in Figure 11 and is divided into web layer, parse layer, domain layer, and util layer. The web layer is responsible for consuming messages from the message queue, downloading and uploading files from S3, and calling corresponding methods in the parse layer based on the message content. The parse layer contains methods for processing business logic specific to each message, calling different scripts based on the message body, and writing the processing results into the database. The domain layer is the entity relationship layer, which stores the database mapping of all back-end entities and

entity message mappings. The util layer is a general-purpose tool package, which includes database connection, storage, message queue, and some general methods.

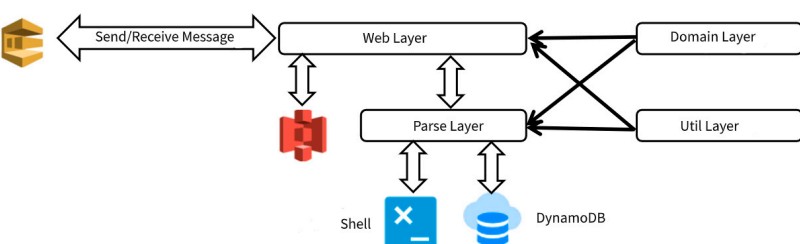

**Figure 11.** Background Structure.

- Web Layer

The method used in this layer is polling. Polling is performed every 10 s without querying the message. Once the message is queried, it is removed from the message queue and processed accordingly based on the message content, as shown in Figure 12. The process is divided into the following steps and instructions:

Firstly, extract message body information. Extract the value of each parameter in the message body for later use.

Secondly, create a server folder. Due to the large size of the computed files and the potential consequences of storing them on the server in case of failure, store all files in S3 and download as needed. Therefore, create a directory similar to the S3 storage structure in the server to download the corresponding files.

Thirdly, verify the file. Even if the existence of the file has been verified before the API sends the message, there is no guarantee that the file is absolutely safe in the process of use. Therefore, the checksum must be performed in the background. Additionally, some calculations need to rely on the calculation results of the previous message queue. If not queried, the system also needs to perform the corresponding processing, such as message call back, throwing exceptions, etc.

Fourthly, download the files. Download the corresponding files from S3 after all checks have been completed, which can take longer if the file is larger.

Fifthly, start the calculation. Choose the appropriate processing method with which to calculate the file and then write the result into the database.

Finally, upload files. Some intermediate files are generated during the computation, which will be used in the following computations, and so they need to be uploaded to S3 for downloading when other computations are started. Since the background server does not store files, all downloaded files and files generated during computation are deleted after all computations are completed in order to reduce the server's hard disk footprint and ensure high file availability.

The entire polling method involves only sending, consuming, and receiving messages, in addition to checking, downloading, and uploading files, and overall does not involve specific business processing logic. All processing is performed using the parse layer processing methods.

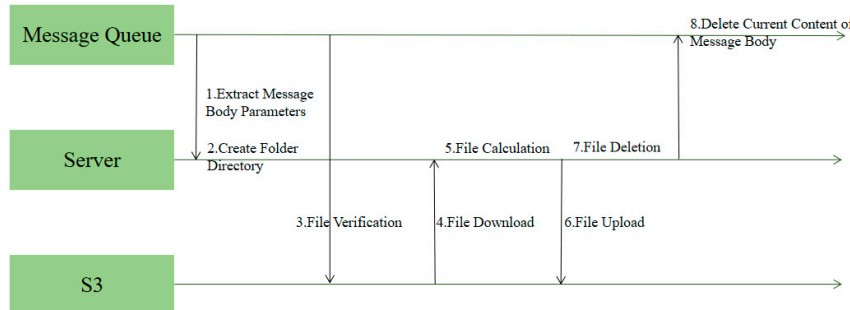

**Figure 12.** Polling Method.

- Parse Layer

The method in this layer is to perform SCADA interpolation. Once all files and parameters have been prepared in the previous step, the next method calls for researchers to start the computation. The computation scripts are written in R and Python and are executed using the command line. All scripts write the results of the computation to multiple CSV files. If an error occurs during the computation, the method throws an exception and handles it accordingly. If the computation is completed successfully, the method reads the corresponding file, organizes all the results in the database format, and writes the data into DynamoDB.

- Domain Layer

The domain layer stores a large number of entity–relationship mappings and message–relationship mappings, some of which are shown in Table 1.

**Table 1.** Domain Entity.

| Name | Introduction |
|---|---|
| Mast Msg Body | Wind tower calculation news |
| NTF Pre Msg Body | NTF pre-processing information |
| RAMA Msg Body | Run analysis and model validation messages |
| Project Info | Project Information |
| SCADA Data Minute | SCADA minute time series |
| SCADA Data summary | SCADA overall statistics |
| Mast Summary | Overall statistics of wind measurement towers |
| Mast Data Minute | Wind measurement tower minute time series |
| Wind Rose | Wind Attack Speed |
| Wind shear | Wind shear |
| NTF Summary | NFT Overall Statistics |
| Terrain Check Summary | Overall statistics of terrain calibration |
| RAMA Summary | Overall statistics for run analysis and model validation |

- Util Layer

The Util layer stores a large number of generic methods that can be called by other modules in order to optimize the code structure and improve code readability. Some examples of method names and functions are shown in Table 2.

**Table 2.** Util Method.

| Method | Introduction |
|---|---|
| Connect DynamoDB | Connecting to DynamoDB Databases |
| Delete Object | Delete the file object on S3 |
| Delete By Folder Prefix | Delete file objects on S3 by prefix matching |
| Upload Or Update | Upload or update files to S3 |
| Get And Del Message | Get and delete messages in SQS |
| Send Message | Sending messages to the SQS |
| Pre-Input | DynamoDB Throughput Pre-Processing |
| Post-Input | DynamoDB Throughput Post-Processing |
| String To Double | String to double with null handling |
| Parse Date Format | Date format standardization |
| Get Fitting Arr By Fitting Code | Returns an array of fit types based on the fit type designator |
| Get Classify Type By Fitting Code | Returns the classification type based on the fit type code |
| Get Wind By Id and Number | Generate standardized unit number based on wind farm id and unit number |

## 5. Test and Discussion

System testing is conducted to evaluate the entire system, including hardware, software, and operators, in order to identify any deviations from the system design. This

type of testing can help to detect errors in system analysis and design, such as whether adequate security measures are in place to prevent illegal intrusions into the system, as well as whether the system can function normally under both normal and overload conditions.

For this system, we have divided testing into two parts: interface testing and message testing. Interface testing focuses mainly on whether all published restful API are available and whether they handle exceptions properly. On the other hand, message testing mainly checks whether messages sent through the API can be processed correctly and whether the system can respond appropriately when errors occur.

This system comprises seven modules with over a hundred API and rich functionalities. During testing, we will focus only on the main API, and will not present the testing of other functionalities here.

- Home API Test Cases
- The test cases for the Home API are shown in Table 3.

**Table 3.** Home Page API Test Cases.

| No. | Test Sub-Items | Implementation Steps | Expected Results | Actual Results |
|---|---|---|---|---|
| 1 | Generate project ID | Pass in the user ID to generate the project ID | Generate a new project ID, in the format of username plus time plus the number of items created on that day | Same as expected results |
| | | After a user has created 99 items in a day, continue passing in the user ID to generate | Failed to create, reached the day's creation limit | Same as expected results |
| 2 | Obtain a list of projects | Pass in the user ID to obtain all the items visible to that user | Search for information on all items that meet the criteria | Same as expected results |
| 3 | Add projects | Enter the item name, item description and item | Create project successfully | Same as expected results |
| 4 | Delete projects | Pass in the ID of the deleted item for item deletion | Pseudo-deletion of items in database successful | Same as expected results |
| 5 | Modify projects | Pass in the modified item information for modification | Modified successfully | Same as expected results |
| 6 | Access to owners and areas | Call the interface to obtain all owners and areas of the business | Obtain json strings of owners and regions successfully | Same as expected results |
| 7 | Access to wind farm information | Pass in the correct wind farm ID and obtain the details of that wind farm | Successful acquisition of wind farm information | Same as expected results |
| | | Obtain wind farm details after passing in a non-existent wind farm ID | No access to wind farm information | Same as expected results |
| 8 | Start SCADA interpolation | SCADA interpolation is carried out after uploading the appropriate files and passing in the parameters for starting the interpolation | Successfully sends the message initiating SCADA interpolation to the appropriate message queue | Same as expected results |
| 9 | Obtain wind speed statistics for all projects | Pass in the user ID and obtain the wind speed statistics for the project | Obtain wind speed statistics for all projects that the user can view | Same as expected results |
| 10 | Obtain full TBA-PBA information for all projects | Obtain the TBA-PBA information of the project after passing in the user ID | Obtain TBA-PBA information for all the projects that the user has access to | Same as expected results |

- Wind Measurement Tower API Test Cases
- An example of a test case for Wind Measurement Tower API is shown in Table 4.

**Table 4.** Mast API Test Cases.

| No. | Test Subsets | Execution Steps | Expected Results | Actual Results |
|---|---|---|---|---|
| 1 | Obtain project wind tower information | Pass in the correct project ID to obtain all the wind tower information | Obtain information on all wind towers for the project | Same as expected results |
| 2 | Obtain basic information about wind measurement towers | Pass in a non-existent project ID to obtain information about all the wind measurement towers | No information available | Same as expected results |
| 3 | Obtain wind shear information | Pass in the tower ID to obtain the wind shear information of the wind tower | Obtain the basic information of the corresponding wind measurement tower | Same as expected results |
| 4 | Obtain monthly average wind speed distribution | Pass in the tower ID to obtain the monthly average distribution information of the wind tower | Obtain the corresponding monthly average distribution information | Same as expected results |

- Message Testing Cases

Message test refers to whether the back-end can perform the corresponding calculations for the messages sent by the API, and whether the corresponding errors can be handled and returned correctly, etc. A total of seven message queues are designed according to the business requirements, and each message queue is tested accordingly, and the test cases are shown in Table 5.

**Table 5.** SQS Test Cases.

| No. | Test Sub-Items | Implementation Steps | Expected Results | Actual Results |
|---|---|---|---|---|
| 1 | Wind measurement towers | Upload the right file and send the right message | Calculations are completed correctly, and the results are written into the database | Same as expected results |
| | | File not uploaded but message sent | Calculation failed, return file not uploaded error | Same as expected results |
| | | Message sent in wrong format or with wrong message content | Calculation failed; message error returned | Same as expected results |
| | | Document content or formatting errors | Calculation failed, an error in the file format returned | Same as expected results |
| 2 | SCADA pre-processing | Upload the right file and send the right message | Calculations are completed correctly, and the results are written into the database | Same as expected results |
| | | File not fully uploaded but message sent | The calculation was partially successful because only a part of the file was uploaded and only a part of the file could be processed | Same as expected results |
| | | Message sent in wrong format or with wrong message content | Calculation failed; message error returned | Same as expected results |
| | | Document content or formatting errors | Calculation failed, an error in the file format returned | Same as expected results |
| 3 | Terrain calibration | Upload the right file and send the right message | Calculations are completed correctly, and the results are written into the database | Same as expected results |
| | | File not uploaded but message sent | Calculation failed, an error that the file did not exist returned | Same as expected results |
| | | Message sent in wrong format or with wrong message content | Calculation failed; message error returned | Same as expected results |
| | | Wrong file content or format, or mismatch between terrain file and coordinate file | Calculation failed, an error in the file format returned | Same as expected results |
| 4 | SCADA interpolation | Upload the right file and send the right message | Calculations are completed correctly, and the results are written into the database | Same as expected results |
| | | File not uploaded but message sent | Calculation failed, an error that the file did not exist returned | Same as expected results |
| | | Message sent in wrong format or with wrong message content | Calculation failed; message error returned | Same as expected results |
| | | Document content or formatting errors | Calculation failed, an error in the file format returned | Same as expected results |
| 5 | NTF pre-processing | Upload the right file and send the right message | Calculations are completed correctly, and the results are written into the database | Same as expected results |
| | | File not uploaded but message sent | Calculation failed, an error that the file did not exist returned | Same as expected results |
| | | Message sent in wrong format or with wrong message content | Calculation failed; message error returned | Same as expected results |
| | | Document content or formatting errors | Calculation failed, an error in the file format returned | Same as expected results |

The test results indicate that the back-end is able to successfully retrieve the necessary parameters and files for computation based on the message content sent by the front-end. The system is able to write the correct results into the database and storage, and handle errors or messages that require waiting in a proper manner, which aligns with the intended design goals and fulfills the system requirements.

## 6. Conclusions

In this paper, we propose an intelligent grid power generation post-evaluation platform based on a micro-service framework and big data analysis in an effort to solve the problems of traditional grid power generation post-evaluation. We use a micro-service architecture to build this platform, which has good scalability and maintainability, and utilize big data analysis technology to visually display and analyze power generation data

from multiple dimensions, thereby improving the accuracy and efficiency of grid power generation post-evaluation.

Through the implementation and testing of the platform, we found that it can quickly and accurately perform post-evaluation of power generation, and provide real-time monitoring and warning services for power companies. The main features of the system are as follows:

- By getting rid of traditional offline mode and client software mode, wind resource engineers no longer need to perform post-assessment calculations manually or install client software, greatly improving the efficiency of post-assessment of wind farm power generation.
- The uniformity of the evaluation method and the determination of evaluation standards are no longer based on personal experience, but are determined by the platform's back-end algorithm, ensuring the professionalism and reliability of the analysis results.
- The results display is clear. Unlike traditional methods that only view massive raw data, the use of a large number of charts and graphs makes the calculation results more obvious and easier to read, and helps to make reasonable judgments about the operation status of the wind farm.
- The iteration speed is improved. Due to the stability of input and output, only continuously iterating the back-end algorithm can improve the accuracy of calculations.
- By developing such an evaluation platform, bold attempts have been made to microservices, big data, and other technologies, providing a solution for the application of big data technology in the smart grid and accumulating experience in developing big data digital platforms in other fields.

However, there were still some problems encountered in the construction of the system platform. For example, using too many programming languages made it difficult to integrate multiple languages and handle exceptions. Additionally, the calls between multiple micro-service modules were not connected. Additionally, since the system platform was designed only for professionals, the demands for multi-user and high-concurrency scenarios have not been considered [44]. Therefore, in future research, we plan to further improve the functionality and performance of the platform, increase the system's concurrency, introduce blockchain technology to improve the system's security, and apply it to more intelligent grid systems to achieve more accurate and secure grid management and optimization.

**Author Contributions:** Methodology, W.W. (Wu Wen) and A.T.; Software, W.W. (Wei Wang); Validation, X.W.; Writing—original draft, J.W. and R.O.; Writing—review & editing, W.W. (Wu Wen) and X.Z. All authors have read and agreed to the published version of the manuscript.

**Funding:** This work was funded by the Researchers Supporting Project number (RSPD2023R681) King Saud University, Riyadh, Saudi Arabia.

**Data Availability Statement:** No public data sets were used to support this study.

**Conflicts of Interest:** The authors declare no conflict of interest.

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
