# Peer review of "A Post-Evaluation System for Smart Grids Based on Microservice Framework and Big Data Analysis"

_electronics, doi:10.3390/electronics12071647_

Round 1

Reviewer 1 Report

In this paper, a post-evaluation system platform for smart grid power generation scenarios based on microservice framework and big data analysis is designed. However, after reviewing this paper, several critical issues that highly affect the understanding of the proposed framework were found. The paper in its current form lacks coherence and unable to deliver its purpose to the reader. Accordingly, major changes are required.

1. Sections 1 and 2 need to be more concise and directed only to demonstrate the main idea of the paper. In the current form, those sections are confusing and do not clearly elaborate the purpose, the gaps, and the necessity of this study. For example, in section 2 (related work), there are many references used that has no relation with the topic of the paper, or they only discuss the topic of big data but without any connection to wind energy or smart grids. Therefore, my suggestions regarding those two sections are as follows:

* Sections 1 and 2 should be merged to be only 1 section.

* This section must be simplified to “only include” the literature related to the main topic of this paper. The unrelated works must be deleted to make this section clearer and more concise. In addition, the authors should pay attention to the flow of the information mentioned in this section. In other words, by reading this section, the readers must understand the necessary background of the paper’s topic, the evolution of the post-evaluation of platforms for smart grids while focusing on the detected gaps, and the contributions provided in the paper.

* The contributions of this paper should be mentioned after discussing the related work. Hence, after merging sections 1 and 2, the main contributions should be placed at the end of the new section 1.     

* The sentence in page 2 (line 51 to line 55) needs to be supported by a reference. There are several sentences and information like that also need to be supported by references, such as AutoGrid: page 4 line 154, Opower: page 4 line 157, The University of California, Los Angeles (UCLA): page 4 lines 160,161, EFD: page 4 line 166, the Danish company Vestas Wind: page 4 line 170, etc. Those sentences should not be used in the paper without references. Please check the whole manuscript for similar issues and fix them.       

* What is DRES? Page 3 line 120? What is the reason of writing the words like this: “the proposed DIstributed Alternating Direction mEthod of”? page 3 line 139. Please check the whole manuscript for similar issues and fix them.

2. Section 3 is relatively long and only very few references were used to support it, which is totally unacceptable. More references should be used in this section. Moreover, it is better to simplify the subsections of this section, especially 3.2.1.

In addition, it is better to use more figures, flowcharts, or tables to demonstrate the presented information of section 3. The presentation of section 3 in the current form is very confusing.

3. Similarly, the demonstration of section 4 should basically rely on figures, flowcharts, or tables. Since this section is for the implementation of the proposed system, and no mathematical models or equations are used, it is much better to increase the understanding and visualization of the proposed method. The presentation of section 4 in the current form is very confusing.

4. The conclusion should be modified to reflect the findings of the paper.

5. The English writing of this paper needs several improvements. An extensive editing is required.    

Author Response

Response to Reviewer 1 Comments

Comment 1: Sections 1 and 2 need to be more concise and directed only to demonstrate the main idea of the paper. In the current form, those sections are confusing and do not clearly elaborate the purpose, the gaps, and the necessity of this study. For example, in section 2 (related work), there are many references used that has no relation with the topic of the paper, or they only discuss the topic of big data but without any connection to wind energy or smart grids. Therefore, my suggestions regarding those two sections are as follows:

* Sections 1 and 2 should be merged to be only 1 section.

* This section must be simplified to “only include” the literature related to the main topic of this paper. The unrelated works must be deleted to make this section clearer and more concise. In addition, the authors should pay attention to the flow of the information mentioned in this section. In other words, by reading this section, the readers must understand the necessary background of the paper’s topic, the evolution of the post-evaluation of platforms for smart grids while focusing on the detected gaps, and the contributions provided in the paper.

* The contributions of this paper should be mentioned after discussing the related work. Hence, after merging sections 1 and 2, the main contributions should be placed at the end of the new section 1.     

* The sentence in page 2 (line 51 to line 55) needs to be supported by a reference. There are several sentences and information like that also need to be supported by references, such as AutoGrid: page 4 line 154, Opower: page 4 line 157, The University of California, Los Angeles (UCLA): page 4 lines 160,161, EFD: page 4 line 166, the Danish company Vestas Wind: page 4 line 170, etc. Those sentences should not be used in the paper without references. Please check the whole manuscript for similar issues and fix them.       

* What is DRES? Page 3 line 120? What is the reason of writing the words like this: “the proposed DIstributed Alternating Direction mEthod of”? page 3 line 139. Please check the whole manuscript for similar issues and fix them.

Response 1:

Thank you for your suggestions and apologize for the problems with our article. We have first improved the English writing throughout the entire text, and then made modifications to the content. We subdivided the first part into three sections, namely "1.1 Motivation", "1.2 Research Challenges", and "1.3 Contributions", to more clearly express our motivation, challenges, and contributions.

In Section 2 Related Work, we have removed redundant content and irrelevant literature, and introduced the evolution process of grid post-evaluation from the manual stage to the intelligent stage, as well as the differences in post-evaluation methods in different stages. Meanwhile, we have made modifications to the sentences that you suggested needed supporting references, and checked the entire manuscript for similar issues. Specifically, we added references in the relevant paragraphs and made adjustments and modifications to the language. The specific additions and modifications are as follows:

  1. Introduction

In recent years, with the growing global demand for energy and increasing awareness of environmental protection, wind energy has become a popular clean energy source. More and more countries and regions are investing in wind farms to meet their energy needs and promote sustainable development. However, as the number of wind farms continues to increase, efficient management and monitoring of these distributed wind power facilities has become a major challenge. These wind farms usually consist of hundreds of wind turbines, each of which needs to be monitored and maintained. In addition, the instability of wind energy also makes wind farm management more challenging, as changes in wind speed and direction can have a significant impact on the performance and output of wind turbines.

In this context, big data technology can play an important role. By monitoring and collecting large amounts of wind farm data, such as wind speed, wind direction, turbine output, and power load, management personnel can better understand the operational status of wind farms. In addition, big data technology can also be used for prediction and optimization of wind farms, such as optimizing turbine output by predicting changes in weather and wind speed, and adjusting the load balance of the power grid.

1.1. Motivation

Post-evaluation of wind farm power generation is the process of evaluating the performance of wind turbines and determining their electricity generation capabilities. With the increasing number and size of wind farms, the demand for power post-evaluation has also been on the rise. In the current era of rapid technological development, it is necessary to continuously apply new technologies to practical business and continuously innovate on this basis to maintain a competitive advantage in the wind power industry. However, wind farm researchers currently rely only on some manual analysis software to evaluate a large amount of wind farm data. As data volumes increase and delay-sensitive applications continue to develop, the need for ubiquitous connectivity and high-accuracy analysis continues to rise, and traditional manual analysis methods have become inefficient. In this context, it is necessary to develop a digital platform for post-evaluation of wind farm power generation based on big data technology to promote the modernization and intelligence of wind farm platforms and complete the overall planning of enterprise wind farms.

1.2. Research Challenge

The continuous development of big data technology has brought new opportunities for wind farm power output assessment. The application of big data technology to post-evaluation of wind farm power generation can enable more precise evaluation of wind turbine performance and output, thereby improving the operational efficiency and power generation capacity of wind farms. However, such assessment typically involves a large amount of data processing and analysis, and thus poses certain challenges.

  • Incomplete or missing data: In wind farm power post-evaluation, collected data may be incomplete or missing, which may reduce the accuracy and reliability of the assessment results.
  • Inconsistent data quality: The data generated by wind turbines during operation may contain noise or errors, which may affect the accuracy of data analysis. In addition, due to the distributed nature of wind turbines, data may come from different sources, which may have different data quality standards.
  • Difficult to establish accurate benchmarks: To determine the performance of wind turbines, it is necessary to compare them with data from other similar turbines. However, due to differences in turbine models, environmental conditions, and service life, establishing accurate benchmarks may be challenging.
  • Difficult to predict future power generation: Environmental factors such as wind speed and direction can affect the performance and output of wind turbines, making it uncertain to evaluate their future power generation capacity.
  • Large-scale of data analysis and processing: Wind farms generate a vast amount of data, which requires a large amount of data analysis and processing to extract valuable information. This requires the use of advanced algorithms and analytical techniques to process and analyze a large volume of data.

1.3. Contributions

This article designs a platform for post-evaluation of wind farm power generation based on big data processing and analysis techniques, as well as distributed software architecture, to address related problems. The main contributions of this paper are as follows:

  • This article uses big data processing technology to improve the efficiency of post-evaluation calculation and processing, and combines it with specific post-evaluation business content of wind farm power generation. At the same time, the use of monitoring and control, data acquisition, and nacelle transfer functions are used to complete the quantitative evaluation of post-evaluation of power generation, thereby ensuring the professionalism and reliability of the post-evaluation.
  • The system combines distributed software architecture, which can significantly improve the overall throughput of the system compared to traditional centralized application systems, reduce system coupling and system latency. This can better achieve efficient operation of the system, and improve the reliability and stability of the entire system.
  • The system applies big data visualization technology to the smart grid, realizing the visualization of massive data processing results. The calculation results after big data processing are more intuitive and readable, which is beneficial for wind resource engineers to make reasonable judgments about the operation status of wind farms, thus better ensuring the operating efficiency and power generation efficiency of the system.
  • This article presents a comprehensive big data-based solution for the construction of a post-evaluation system for power generation in wind farms. It provides useful ideas and references for the future construction of post-evaluation platforms for power generation based on big data.

The remaining sections of this article are organized as follows: Section 2 summarizes related research on smart grids and their associated areas. Section 3 introduces the system design from the perspective of post-evaluation of wind resources in wind farms. Section 4 presents the implementation of the system. Section 5 discusses the system testing and presents the test results. Section 6 provides a discussion and summary of this article.

Comment 2:  Section 3 is relatively long and only very few references were used to support it, which is totally unacceptable. More references should be used in this section. Moreover, it is better to simplify the subsections of this section, especially 3.2.1.

In addition, it is better to use more figures, flowcharts, or tables to demonstrate the presented information of section 3. The presentation of section 3 in the current form is very confusing.

Response 2:

Thanks for your suggestions. In Section 3, we simplified the module details in Section 3.1 and Sections 3.2.1 and 3.2.3, and added the corresponding references. The specific contents  are as follows:

To achieve front-end and back-end separation and decoupling, the system adopts a front-end and back-end separation architecture pattern. The front-end uses h5[33], Echart[34], and Node.js technology[35] to access the system, transmit data, and display, beautify, and render results by accessing predefined API interfaces. To ensure the stability and invariability of the interfaces, all API developed by the back-end are forwarded by NGINX[36]. All requests are received and processed by the gateway layer, and only requests that meet the requirements are sent to the API layer for access and calculation. To ensure the decoupling and independent development of modules, each module is a service, and all services are deployed on Docker[37] for ease of version iteration and redeployment.

For most API, they only involve data creation, retrieval, updating, and deletion, and they directly access DynamoDB[38] for corresponding operations. Some API involve file transmission and computation, and these API need to store files in Simple Storage Service (S3) or send messages to Simple Storage Service (SQS)[39]. The back-end receives and consumes these messages, and launches different calculation scripts according to the type and content of the message, finally writes the calculation results to DynamoDB and S3, and notifies the system when the computation is completed.

References

  1. Zhang, Y and Cao, S. "Based panoramic technology development and application of H5," 2016 2nd IEEE International Conference on Computer and Communications (ICCC) 2016, pp. 761-764. https://doi: 1109/CompComm.2016.7924805.
  2. Zeng, X.; Zhong, X.; E, Z.; Luo, M.; Zhang, X. Research on data visualization design of poverty alleviation achievements based on echarts. CIBDA 2022; 3rd International Conference on Computer Information and Big Data Applications 2022, pp. 1-6.
  3. Sterling, A. NodeJS and Angular Tools for JSON-LD. 2019 IEEE 13th International Conference on Semantic Computing (ICSC), 2019, pp. 392-395. https://doi: 10.1109/ICOSC.2019.8665625.
  4. Vujović, M.; Savić, M; Stefanović, D; Pap, I. USAGE OF NGINX and websocket in IoT. 2015 23rd Telecommunications Forum Telfor (TELFOR) 2015, pp. 289-292. https://doi: 10.1109/TELFOR.2015.7377467.
  5. Krasnov, A.; Maiti, R. R.; Wilborne, D. M. Data Storage Security in Docker. 2020 SoutheastCon 2020, pp. 1-1. https://doi: 10.1109/SoutheastCon44009.2020.9249757.
  6. Kalid, S.; Syed, A.; Mohammad, A.; Halgamuge, M. N. Big-data NoSQL databases: A comparison and analysis of “Big-Table”, “DynamoDB”, and “Cassandra”. 2017 IEEE 2nd International Conference on Big Data Analysis (ICBDA) 2017, pp. 89-93. https://doi: 10.1109/ICBDA.2017.8078782.
  7. Yoon, H.; Gavrilovska, A.; Schwan, K.; Donahue, J. Interactive Use of Cloud Services: Amazon SQS and S3. 2012 12th IEEE/ACM International Symposium on Cluster, Cloud and Grid Computing (ccgrid 2012) 2012, pp. 523-530. https://doi: 10.1109/CCGrid.2012.85.

Comment 3: Similarly, the demonstration of section 4 should basically rely on figures, flowcharts, or tables. Since this section is for the implementation of the proposed system, and no mathematical models or equations are used, it is much better to increase the understanding and visualization of the proposed method. The presentation of section 4 in the current form is very confusing.

Response 3:

Thank you for your constructive suggestion. In the fourth section, we have added visual charts corresponding to each module of the system. Additionally, we have added a flowchart for the back-end Web Layer in section 4.2 to clearly illustrate the polling process. Below are the visual interface diagrams for each module of the system that we have added.

Figure 4. Project Module

Figure 5. Detection Control and Data Acquisition Module

Figure 6. Wind Measurement Tower Module

Figure 6. Cabin Transfer Function Module

Figure 7. Operation Analysis Module

Figure 8. Model Validation Module

Figure 9. Report module

Figure 11. Polling Method

Comment 4: The conclusion should be modified to reflect the findings of the paper.

Response 4:

Thank you for your suggestion. We have revised and refined the conclusion of this text. The specific modifications are as follows:

In this paper, we propose an intelligent grid power generation post-evaluation platform based on a micro-service framework and big data analysis, aiming to solve the problems of traditional grid power generation post-evaluation. We use a micro-service architecture to build this platform, which has good scalability and maintainability, and utilize big data analysis technology to visually display and analyze power generation data from multiple dimensions, thereby improving the accuracy and efficiency of grid power generation post-evaluation.

Through the implementation and testing of the platform, we found that it can quickly and accurately perform post-evaluation of power generation, and provide real-time monitoring and warning services for power companies. The main features of the system are as follows:

  • By getting rid of traditional offline mode and client software mode, wind resource engineers no longer need to perform post-assessment calculations manually or install client software, greatly improving the efficiency of post-assessment of wind farm power generation.
  • The uniformity of the evaluation method and the determination of evaluation standards are no longer based on personal experience, but are determined by the platform's backend algorithm, ensuring the professionalism and reliability of the analysis results.
  • The results display is clear. Unlike traditional methods that only view massive raw data, the use of a large number of charts and graphs makes the calculation results more obvious, easy to read, and helps to make reasonable judgments about the operation status of the wind farm.
  • The iteration speed is improved. Due to the stability of input and output, only continuously iterating the backend algorithm can improve the accuracy of calculations.
  • By developing such an evaluation platform, bold attempts have been made to microservices, big data, and other technologies, providing a solution for the application of big data technology in the smart grid and accumulating experience for the development of big data digital platforms in other fields.

However, there were still some problems encountered in the construction of the system platform. For example, using too many programming languages made it difficult to integrate multiple languages and handle exceptions. And the calls between multiple micro-service modules were not connected. Additionally, since the system platform was designed only for professionals, the demands for multi-user and high-concurrency scenarios have not been considered. Therefore, in future research, we plan to further improve the functionality and performance of the platform, increase the system's concurrency, introduce blockchain technology to improve the system's security, and apply it to more intelligent grid systems to achieve more accurate and secure grid management and optimization.

Comment 5: The English writing of this paper needs several improvements. An extensive editing is required.

Response 5: Thanks for the suggestion. we have invited a native speaker to help us polish  this paper.

Reviewer 2 Report

Research problems

The actual research problem needs to be discussed clearly. Why this research is so important? Why this research is better than previous research that leads you to conduct this research. There must be research gaps and limitations in between. 

Conclusion:

The following items are not clear. You need to revise this section.

What are the actual contributions especially in terms of theoritical, managerial and practical contributions?

What is/ are the limitation of this study?

What is the future research?

Author Response

Response to Reviewer 2 Comments

Comment 1: Research problems

The actual research problem needs to be discussed clearly. Why this research is so important? Why this research is better than previous research that leads you to conduct this research. There must be research gaps and limitations in between.

Response 1:

Thank you for your suggestions. We have made revisions to the first section of our paper, specifically in section 1.1 where we discuss the importance of our research work. Additionally, in sections 1.2 Research Challenges and 1.3 Contributions, we address the existing challenges in the current research and elaborate on the solutions and contributions that our paper provides in order to tackle these challenges.

Comment 2: Conclusion

The following items are not clear. You need to revise this section.

  • What are the actual contributions especially in terms of theoritical, managerial and practical contributions?
  • What is/are the limitation of this study?
  • What is the future research?

Response 2:

Thank you for your suggestions. In the new version of our paper, we have re-summarized our practical contributions and discussed the limitations of current research, as well as future research directions. Specifically, they are as follows:

In this paper, we propose an intelligent grid power generation post-evaluation platform based on a micro-service framework and big data analysis, aiming to solve the problems of traditional grid power generation post-evaluation. We use a micro-service architecture to build this platform, which has good scalability and maintainability, and utilize big data analysis technology to visually display and analyze power generation data from multiple dimensions, thereby improving the accuracy and efficiency of grid power generation post-evaluation.

Through the implementation and testing of the platform, we found that it can quickly and accurately perform post-evaluation of power generation, and provide real-time monitoring and warning services for power companies. The main features of the system are as follows:

  • By getting rid of traditional offline mode and client software mode, wind resource engineers no longer need to perform post-assessment calculations manually or install client software, greatly improving the efficiency of post-assessment of wind farm power generation.
  • The uniformity of the evaluation method and the determination of evaluation standards are no longer based on personal experience, but are determined by the platform's backend algorithm, ensuring the professionalism and reliability of the analysis results.
  • The results display is clear. Unlike traditional methods that only view massive raw data, the use of a large number of charts and graphs makes the calculation results more obvious, easy to read, and helps to make reasonable judgments about the operation status of the wind farm.
  • The iteration speed is improved. Due to the stability of input and output, only continuously iterating the backend algorithm can improve the accuracy of calculations.
  • By developing such an evaluation platform, bold attempts have been made to microservices, big data, and other technologies, providing a solution for the application of big data technology in the smart grid and accumulating experience for the development of big data digital platforms in other fields.

However, there were still some problems encountered in the construction of the system platform. For example, using too many programming languages made it difficult to integrate multiple languages and handle exceptions. And the calls between multiple micro-service modules were not connected. Additionally, since the system platform was designed only for professionals, the demands for multi-user and high-concurrency scenarios have not been considered. Therefore, in future research, we plan to further improve the functionality and performance of the platform, increase the system's concurrency, introduce blockchain technology to improve the system's security, and apply it to more intelligent grid systems to achieve more accurate and secure grid management and optimization.

Round 2

Reviewer 1 Report

1. The added references are all conference papers. Why is that? In fact, the majority of this paper's references are now conference papers. It is suggested to replace several of them with journal references.  

2. The number of the added references as mentioned in the author's response is different from their numbers in the reference list, which is totally wrong and unacceptable. Please check the whole reference list carefully.  

3. The paper's format and language is still poor and need more improvement. 

4. Some suggested comments in my first report were totally ignored without any response. Please read the first-round report carefully again. Regarding the comments that are unable to reply, please provide required explanations. Moreover, it was not easy to find the modifications and changes made by the authors in the revised manuscript. For example, many changes were made in sections 3 and 4 without showing that in the revised manuscript. As a kind suggestion, please be more careful writing the response as well as demonstrating the modified parts next time.     

Author Response

Dear Editor and Reviewers,

Thank you again for insightful corrections and time spent on this paper, which greatly help us to improve the paper’s quality. We have made effort to modify this paper according to the reviewers’ comments. The details of our responses are shown in below.

Sincerely,

Authors

Response to Reviewer 1’s Comments

Comment 1: The added references are all conference papers. Why is that? In fact, the majority of this paper's references are now conference papers. It is suggested to replace several of them with journal references. 

Response 1: Thank you very much for your suggestion, we are very sorry for the many problems in our paper, as you said, we have a large number of conference papers in our paper, for this reason we did a lot of journals reading and replaced some of the conference papers, the replacement journals are as follows. Since an additional reference was added [18], the reference numbers are not the same as the original after replacement.

Original References

  1. Barai, G. R. ; Krishnan, S.; Venkatesh, B. Smart metering and functionalities of smart meters in smart grid - a review. 2015 IEEE Electrical Power and Energy Conference (EPEC) 2015, pp. 138-145. https://doi: 10.1109/EPEC.2015.7379940.
  2. Sanchez A.; Rivera W. Big Data Analysis and Visualization for the Smart Grid. 2017 IEEE International Congress on Big Data (BigData Congress) 2017, pp. 414-418. https://doi: 10.1109/BigDataCongress.2017.59.
  3. Hou, L.; Zhang, Y.; Yu, Y.; Shi, Y.; Liang, K. Overview of Data Mining and Visual Analytics towards Big Data in Smart Grid. 2016 International Conference on Identification, Information and Knowledge in the Internet of Things (IIKI) 2016, pp. 453-456. https://doi: 10.1109/IIKI.2016.83.
  4. Xu, Y.; Zhang, J.; Wang, W.; Juneja, A.; Bhattacharya, S. Energy router: Architectures and functionalities toward Energy Internet. 2011 IEEE International Conference on Smart Grid Communications (SmartGridComm) 2011. https://doi.org/10.1109/SmartGridComm.2011.6102340.
  5. Refaat, S. S.; Mohamed, A.; Abu-Rub, H. Big data impact on stability and reliability improvement of smart grid. 2017 IEEE International Conference on Big Data (Big Data), 2017, 1975-1982. https://doi.org/10.1109/BigData.2017.8258143
  6. Goel, P.; Datta, A.; Mannan, M. S. Application of big data analytics in process safety and risk management. 2017 IEEE International Conference on Big Data (Big Data), 2017, 1143-1152. https://doi.org/10.1109/BigData.2017.8258040.
  7. Zhang, Y. S.; Cao, S. X. Based panoramic technology development and application of H5. 2016 2nd IEEE International Conference on Computer and Communications (ICCC) 2016, pp. 761-764. https://doi: 1109/CompComm.2016.7924805.
  8. Vujović, M.; Savić, M; Stefanović, D; Pap, I. USAGE OF NGINX and websocket in IoT. 2015 23rd Telecommunications Forum Telfor (TELFOR) 2015, pp. 289-292. https://doi: 10.1109/TELFOR.2015.7377467.
  9. Kalid, S.; Syed, A.; Mohammad, A.; Halgamuge, M. N. Big-data NoSQL databases: A comparison and analysis of “Big-Table”, “DynamoDB”, and “Cassandra”. 2017 IEEE 2nd International Conference on Big Data Analysis (ICBDA), 2017, pp. 89-93. https://doi: 10.1109/ICBDA.2017.8078782.
  10. Yoon, H.; Gavrilovska, A.; Schwan, K.; Donahue, J. Interactive Use of Cloud Services: Amazon SQS and S3. 2012 12th IEEE/ACM International Symposium on Cluster, Cloud and Grid Computing (ccgrid 2012) 2012, pp. 523-530. https://doi: 10.1109/CCGrid.2012.85.

Replace References

  1. Miao, H.; Chen, G.; Zhao, Z.; Zhang, F. Evolutionary Aggregation Approach for Multihop Energy Metering in Smart Grid for Residential Energy Management. IEEE Transactions on Industrial Informatics 2021, 17, 1058-1068. https://doi.org/10.1109/TII.2020.3007318.
  2. Chung, H. -M.; Maharjan, S.; Zhang, Y.; Eliassen, F. Distributed Deep Reinforcement Learning for Intelligent Load Scheduling in Residential Smart Grids. IEEE Transactions on Industrial Informatics 2021, 17, 2752-2763. https://doi.org/10.1109/TII.2020.3007167.
  3. Su, Z. et al., Secure and Efficient Federated Learning for Smart Grid With Edge-Cloud Collaboration. IEEE Transactions on Industrial Informatics 2022, 18, 1333-1344. https://doi.org/10.1109/TII.2021.3095506.
  4. Liang, H.; Hua, H.; Qin, Y.; Ye, M.; Zhang, S.; Cao, J. Stochastic Optimal Energy Storage Management    for Energy Routers Via Compressive Sensing. IEEE Transactions on Industrial Informatics 2022, 18, 2192-2202. https://doi.org/10.1109/TII.2021.3095141.
  5. Ketter, W.; Collins, J.; Saar-Tsechansky, M.; Marom, O. Information Systems for a Smart Electricity Grid: Emerging Challenges and Opportunities. ACM Trans. Manage. Inf. Syst, 2018, 9, 22.         https://doi.org/10.1145/3230712.
  6. Bansal, M.; Chana, I.; Clarke, S. A Survey on IoT Big Data: Current Status, 13 V’s Challenges, and Future Directions. ACM Comput. Surv, 2020, 53, 59. https://doi.org/10.1145/3419634.
  7. Wang, L.; Wang, H.; Xue, B.; Zhou, M. H5-Bridge-Based Single-Input–Dual-Output LLC Converter With Wide Output Voltage Range. IEEE Transactions on Industrial Electronics 2022, 69, 7008-7018. https://doi.org/10.1109/TIE.2021.3097597.
  8. Wang, Z.; Lin, J.; Cai, Q.; Wang, Q.; Zha D.; Jing J. Blockchain-Based Certificate Transparency and Revocation Transparency. IEEE Transactions on Dependable and Secure Computing 2022, 19, 681-697.      https://doi.org/10.1109/TDSC.2020.2983022.
  9. Sivasubramanian, S. Amazon dynamoDB: a seamlessly scalable non-relational database service. Proceedings of the 2012 ACM SIGMOD International Conference on Management of Data,         Association for Computing Machinery, 2012, 729-730. https://doi.org/10.1145/2213836.2213945.
  10. Sadeghi, A.; Sheikholeslami, F.; Marques, A. G.; Giannakis, G. B. Reinforcement Learning for Adaptive Caching With Dynamic Storage Pricing. IEEE Journal on Selected Areas in Communications 2019, 37, 2267-2281. https://doi.org/10.1109/JSAC.2019.2933780

New Reference

  1. Wang, X.; Li, J.; Ning, Z.; Song, Q. ; Guo, L.; Guo, S.; Obaidat, M. Wireless Powered Mobile Edge Computing Networks: A Survey. ACM Computing Surveys, 2023. https://doi.org/10.1145/3579992, 2023.

Comment 2The number of the added references as mentioned in the author's response is different from their numbers in the reference list, which is totally wrong and unacceptable. Please check the whole reference list carefully. 

Response 2: We are very sorry for such a low-level error due to our carelessness in not updating the reference numbers in the reply written to you in time, for which we have checked the references in the full text as well as the numbers. The following is the correct version:

To achieve the separation and decoupling of the front-end and back-end, the system adopts a front-end and back-end separation architecture pattern. The front-end uses technologies such as h5 [36], Echart [37], and Node.js [38] to access the system, transmit data, and display, beautify, and render results by accessing predefined API interfaces. To ensure the stability and consistency of the interfaces, all API developed by the back-end are forwarded by NGINX [39]. All requests are received and processed by the gateway layer, and only requests that meet the requirements are sent to the API layer for access and calculation. To ensure the decoupling and independent development of modules, each module is a service, and all services are deployed on Docker [40] for ease of version iteration and redeployment.

For most API, they only involve data creation, retrieval, updating, and deletion, and they directly access DynamoDB [41] for the corresponding operations. Some API involve file transmission and computation, and these API need to store files in Simple Storage Service (S3) or send messages to Simple Queue Service (SQS) [42]. The back-end receives and consumes these messages, and launches different calculation scripts according to the type and content of the message, finally writing the calculation results to DynamoDB and S3, and notifying the system when the computation is completed.

References

  1. Wang, L.; Wang, H.; Xue, B.; Zhou, M. H5-Bridge-Based Single-Input–Dual-Output LLC Converter With Wide Output Voltage Range. IEEE Transactions on Industrial Electronics, 2022, 69, 7008-7018. https://doi.org/10.1109/TIE.2021.3097597.
  2. Zeng, X.; Zhong, X.; E, Z.; Luo, M.; Zhang, X. Research on data visualization design of poverty alleviation achievements based on echarts. CIBDA 2022; 3rd International Conference on Computer Information and Big Data Applications, 2022, pp. 1-6.
  3. Sterling, A. NodeJS and Angular Tools for JSON-LD. 2019 IEEE 13th International Conference on Semantic Computing (ICSC), 2019, pp. 392-395. https://doi: 10.1109/ICOSC.2019.8665625.
  4. Wang, Z.; Lin, J.; Cai, Q.; Wang, Q.; Zha D.; Jing J. Blockchain-Based Certificate Transparency and Revocation Transparency. IEEE Transactions on Dependable and Secure Computing, 2022, 19, 681-697. https://doi.org/10.1109/TDSC.2020.2983022.
  5. Krasnov, A.; Maiti, R. R.; Wilborne, D. M. Data Storage Security in Docker. 2020 SoutheastCon, 2020, pp. 1-1. https://doi: 10.1109/SoutheastCon44009.2020.9249757.
  6. Sivasubramanian, S. Amazon dynamoDB: a seamlessly scalable non-relational database service. Proceedings of the 2012 ACM SIGMOD International Conference on Management of Data, Association for Computing Machinery, 2012, 729-730. https://doi.org/10.1145/2213836.2213945.
  7. Sadeghi, A.; Sheikholeslami, F.; Marques, A. G.; Giannakis, G. B. Reinforcement Learning for Adaptive Caching With Dynamic Storage Pricing. IEEE Journal on Selected Areas in Communications, 2019, 37, 2267-2281. https://doi.org/10.1109/JSAC.2019.2933780.

Comment 3The paper's format and language is still poor and need more improvement. 

Response 3: We are sorry that our writing grammar and formatting problems have brought you a bad experience, we have checked the grammar and formatting of the whole text and made improvements accordingly.

Comment 4Some suggested comments in my first report were totally ignored without any response. Please read the first-round report carefully again. Regarding the comments that are unable to reply, please provide required explanations. Moreover, it was not easy to find the modifications and changes made by the authors in the revised manuscript. For example, many changes were made in sections 3 and 4 without showing that in the revised ma nuscript. As a kind suggestion, please be more careful writing the response as well as demonstrating the modified parts next time. 

Response 4: Once again, we apologize for our first response to you and apologize for overlooking some of your questions due to rough presentation.

For comment ”The sentence in page 2 (line 51 to line 55) needs to be supported by a reference. There are several sentences and information like that also need to be supported by references, such as AutoGrid: page 4 line 154, Opower: page 4 line 157, The University of California, Los Angeles (UCLA): page 4 lines 160,161, EFD: page 4 line 166, the Danish company Vestas Wind: page 4 line 170, etc. Those sentences should not be used in the paper without references. Please check the whole manuscript for similar issues and fix them.   

We have made substantial changes to the first section and removed the parts that were not particularly necessary, including page 2 (line 51 to line 55). The origin of these sentences like AutoGrid and EFD are in the news or some information released by the official websites of related companies, mainly used to introduce some application examples of big data for power grids, but we did not find better references to support it, so this part was deleted.

For comment “What is DRES? Page 3 line 120? What is the reason of writing the words like this: “the proposed DIstributed Alternating Direction mEthod of”? page 3 line 139. Please check the whole manuscript for similar issues and fix them.”

We apologize for the misspelling of words, and we have double-checked the entire text and corrected the errors. DRES is distributed development of renewable energy sources, and the sentence in which the word appears was later modified to ” Lv et al. combined different deep learning algorithms with edge computing to analyze and process the distributed renewable energy generation and consumer power data in intelligent micro-grids, thereby improving information transmission and processing efficiency in the power system” at the end of the third paragraph in Section 2 Related Work. The words "DIADEM, SCADA" at the last paragraph in Section 2 Related Work are also modified.

For comment ”Section 3 is relatively long and only very few references were used to support it, which is totally unacceptable. More references should be used in this section. Moreover, it is better to simplify the subsections of this section, especially 3.2.1. In addition, it is better to use more figures, flowcharts, or tables to demonstrate the presented information of section 3. The presentation of section 3 in the current form is very confusing.”

In addition to adding references in Section 3, we have streamlined this section, and in Section 3.2.1, we simplified some statements, such as ”The project list shows the project name, project address, owner information, installed capacity, model, project founder and other information. Project creation includes project information entry, such as project name, wind farm number, capacity, address, owner, and wind farm operation data upload. The project overview is the view and display of important indicators after the completion of the wind farm assessment, such as the basic information of the wind farm, the operation index of the wind farm, the parameters of the wind tower, and the design information of the wind farm.” simplify to ”The project list displays information such as project name, address, owner information, installed capacity, turbine model, project creator, etc. The project creation module allows users to input project information and upload wind farm operation data. The project overview module displays important indicators after the wind farm assessment is completed.”

Furthermore we have simplified the introduction of SCADA (Supervisory Control And Data Acquisition) module, NTF (Nacelle Transfer Function) module and model verification module in 3.2.1. Function Module Design. In Section 3.2.3, we only introduced the names of the seven message queues, removing the content of the detailed description of each queue. section 3.2.4 is the database design, and this section seemed redundant, so we removed it. section 4.1, we added visual graphical presentations for each module, and in section 4.2, we added references and graphs, such as ”These calculations involve diverse file formats and complex logical judgments, requiring scripts written in languages such as Python and R [42]” and ”The method used in this layer is polling. Polling is performed every 10 seconds without querying the message, and once the message is queried, it is removed from the message queue and processed accordingly based on the message content, as shown in Figure 11.”